

# Adoption of a biologically-enhanced agricultural management (BEAM) approach in agroecosystems for regenerating soil fertility, improving farm profitability and achieving productive utilization of atmospheric $CO_2$

David C. Johnson and Hui-Chun Su Johnson

David C. Johnson LLC, Mesilla, New Mexico, United States

Corresponding authors
David C. Johnson,
johnsoda@nmsu.edu
Hui-Chun Su Johnson,
johnson.su.bioreactor@gmail.com

## ABSTRACT

**Background:** A 4-year field study, on the adoption of a Biologically-Enhanced Agricultural Management (BEAM) protocol, in a cotton/cover-crop rotation in Turkey, was designed to observe "change-over-time" of soil organic carbon (SOC%) and total soil nitrogen (TSN%) at three soil profile depths (0–15 cm, 15–30 cm and 30–45 cm) while tracking farm productivity and profitability.

**Methods:** BEAM systems employ regenerative practices: (a) no-till, (b) no, or reduced synthetic nutrient amendments, (c) continuous roots in the ground (commodity/cover), accompanied with an injection (in-furrow at planting) of an extract of beneficial microbes, from a Johnson-Su bioreactor. Three field nitrogen treatments: (1) BEAM+100% N (203 kg N ha$^{-1}$); (2) BEAM+15% N (30.53 kg N ha$^{-1}$); and (3) BEAM-0% N (No N applied), were implemented, on a 5.22-hectare plot, to assess the influence of BEAM protocols and nitrogen amendments, on SOC %, TSN%, cotton production, and profitability.

**Results:** The SOC%, in the 0–15 cm soil profile demonstrated a significant increase from 0.39% SOC to 1.83% SOC, for a total increase of 1.44%, over the 4-year study period, (y = 0.3136x + 0.1206; r$^2$ = 0.96; F(1,2) = 45.1616, p = 0.02143); The 15–30 cm soil profile demonstrated a non-significant loss of −0.23% SOC (y = −0.3161x + 0.156; r$^2$ = 0.3183; F(1,2) = 0.9339, p = 0.4358), and the 30–45 cm soil profile exhibited a significant increase of 0.28% SOC; (y = 0.0477x + 0.4743; r$^2$ = 0.9363; F(1,2) = 29.4005, p = 0.03237). Annual SOC cumulative increases of ~6.59 metric tons (t) carbon (C) ha$^{-1}$yr$^{-1}$, were observed, from 2019 to 2023, in the top 45 cm of the soil profile along with annual TSN increases of ~0.68 t N ha$^{-1}$yr$^{-1}$ in all three treatments. Cover-crop aboveground biomass increased annually in 2021, 2022 and 2023 from ~400 g, to ~692 g, to ~925 g dry biomass m$^{-2}$yr$^{-1}$ providing annual agroecosystem surface carbon accumulation of ~1.78 t C, ~3.08 t C and ~4.11 t C ha$^{-1}$. Earthworm populations increased from zero earthworms m$^{-2}$ in 2019 to ~100 earthworms m$^{-2}$ in 2023. BEAM protocols also promoted: (a) farm input reductions of: 100% for herbicide, 56% for insecticide, 61% for diesel fuel, 85% for synthetic nitrogen fertilizer, and 100% for phosphorus fertilizer applications, reducing farm input costs ~\$470 ha$^{-1}$yr$^{-1}$. Adoption of a BEAM regenerative agricultural

management system, increased: (a) SOC ($\sim$6.59 t C ha$^{-1}$yr$^{-1}$); (b) C in the annual growth of cover-crop biomass ($\sim$4.12 t of C ha$^{-1}$yr$^{-1}$); (c) carbon in residual surface cover-crop plant residues from previous annual cover crops ($\sim$0.82 t C ha$^{-1}$yr$^{-1}$); and (d) C exported in cotton lint ($\sim$0.77 t C ha$^{-1}$yr$^{-1}$). Total C avoidance included: (a) reductions in fertilizer, pesticides and diesel inputs ($\sim$0.33 t C ha$^{-1}$yr$^{-1}$); and (b) reduction of C respiration from adoption of zero-till ($\sim$0.64 t C ha$^{-1}$yr$^{-1}$). Adoption of BEAM management, in this cotton/cover-crop agroecosystem, provided productive utilization, or avoidance of $\sim$13.27 t of atmospheric C ha$^{-1}$yr$^{-1}$.

## INTRODUCTION

The concept of regenerative agriculture (RA) is still in development due to multiple perspectives from which many practitioners, promoters and researchers observe and interpret applications and outcomes. Regenerative Agriculture, at the core, is an approach to agriculture designed to more closely mimic natural ecosystems (prairies and savannahs) (*Paustian et al., 2020*) to include practices that: (a) reduce soil physical disturbance (*i.e.*, ripping, plowing discing), (b) avoid or reduce application of synthetic nutrients (N, P, K), herbicides and pesticides, (c) employ continuous vegetative cover (either commodity or cover-crops) with continuous roots in the ground and (d) increase system plant, insect, soil fauna and microbiome diversity.

Regenerative agricultural practices are designed to increase the rate, and quantity of carbon flowing into soils, while reducing the rate of carbon loss from soil respiration and erosion (*Paustian et al., 2020*). Promoting efforts to increase soil organic carbon (SOC) in agroecosystems (crop and grazing lands) has been proposed to restore soil fertility, productivity and function (*Garcia, Nannipieri & Hernandez, 2018*), and as a viable method for reducing atmospheric $CO_2$ for over two decades; however, there are multiple obstacles (cultural, policy, economic and physical barriers) that have arisen, regarding the use of soils as a major C sink (*Amundson & Biardeau, 2018*). These obstacles need to be addressed, but the questions that are more relevant now and should be undertaken initially are: can we capture significant amounts of atmospheric $CO_2$, as SOC in agricultural soils, and at what cost to farmers and society?

Many of the regenerative principles have formed as a direct result of the realization that our current agricultural model is unsustainable (*Araújo et al., 2023*). Conventional management practices are promoting multiple destructive impacts to our agroecosystems and the environment including: (a) loss of soil C reserves from excessive soil disturbance and tilling (*Haddaway et al., 2017*); (b) degradation of soils from wind and water erosion, depleting native stocks of topsoil with soil loss rates greater than those of topsoil replacement, potentially limiting the lifespan of agricultural systems (*Montgomery, 2007*); (c) negative plant nutrient balances in the soil from extractive farming practices (*Lal, 2009*); (d) deficiency of essential nutrients, minerals and proteins in our foods, when

compared to food products grown just a few decades earlier (*Scrimshaw & San Giovanni, 1997*; *Mustafa, Mabhaudhi & Massawe, 2021*) and (e) pollution of soils, atmosphere, aquifers, rivers, estuaries and oceans from applications of synthetic fertilizer and pesticides (*Rivett et al., 2008*).

Application of synthetic fertilizers (nitrogen and phosphorus), as nutrient amendments, paired with conventional methods for cultivation of soils has: (a) reduced beneficial associations between plants and soil microbial communities (SMC) (*Mäder et al., 2000*; *Penton et al., 2014*), damaging associated SMC that support plant nutrient acquisition and plant pathogen resistance (*Doornbos, van Loon & Bakker, 2012*); (b) decreased biological activity, and the formation of micro- and macro-aggregates, which provide an important habitat for microbial activity (*Dick, 1992*); (c) threatened soil invertebrate populations (earthworms, nematodes, protists, *etc.*) affecting nutrient cycling, soil structure maintenance, carbon transformation, and regulation of pests and diseases (*Gunstone et al., 2021*), (d) promoted negative impacts on biodiversity, including plant vertebrate and non-vertebrate groups (*McLaughlin & Mineau, 1995*), (e) degraded soil fungal populations through reduced incorporation of fresh plant C (*Dighton, 2003*), and (f) decreased soil fertility through reduction of soil C and microbially-originated C stocks, escalating soil C decomposition rates and disrupting soil microbial food webs (*Huber et al., 2008*; *Horrigan, Lawrence & Walker, 2002*; *Kuzykov, 2010*).

Considering these challenges, there is a growing need for adopting agricultural practices that can begin to restore SMC health and function, provide essential plant nutrients for improving crop productivity and quality, while also promoting significant increases in stable SOC. Recent efforts, to determine the potential for increasing SOC in agroecosystems, towards the mitigation of anthropogenic $CO_2$ emissions, have implemented: (a) conservation no-till, (b) cover-cropping, (c) cover-crop and crop residue mulch, and (d) nutrient recycling through utilization of manure or compost.

Conversion from plough to no-till demonstrated average C accumulation rates of: ~0.570 ± 0.140 t C ha$^{-1}$ yr$^{-1}$ in sixty-seven long-term field experiments (*West & Post, 2002*); (b) 0.48 t to 0.52 t C ha$^{-1}$ yr$^{-1}$ in a meta-analysis of no-till systems (*Porwollik et al., 2022*); and (c) an estimated rate of 0.2 t C ha$^{-1}$ yr$^{-1}$ observed in arable and permanent cropping systems of the world (*Niggli et al., 2009*).

Adoption of no-till practices, plus cover-cropping, for increasing SOC and associated soil health and fertility, have provided mixed to limited increases in three global meta-analyses of multi-year cover-cropping in offseason plantings, demonstrating: (a) ~0.44 t C ha$^{-1}$ yr$^{-1}$ (*Ruis & Blanco-Canqui, 2017*); ~0.56 t C ha$^{-1}$ yr$^{-1}$ (*Jian et al., 2021*) and ~0.32 ± 0.08 t C ha$^{-1}$ yr$^{-1}$ (*Poeplau & Don, 2015*). A meta-analysis of 150 studies considering mulching, determined crop yield was improved ~48%; however, there was no significant effect for promoting SOC increases (*Yu et al., 2021*). The adoption of these regenerative practices, for no-till, cover-cropping, crop residue management and green manuring are beneficial, but they have produced limited effects only increasing SOC from 0.2 to 0.71 t C ha$^{-1}$ yr$^{-1}$ in agroecosystems.

Agricultural land-use intensification (*i.e.*, chemical fertilization, pesticides, *etc.*) has contributed to a reduction in the complexity of the soil microbiome structure as well as a

decrease in the microbial community-weighted mean body mass of soil fauna (*Tsiafouli et al., 2015*). In response to these observations, the application of beneficial microbes, as biofertilizers, are now being considered as an alternative to "intensification" due to observations that "imported" soil microbes may have the ability to enhance crop production and food safety (*Mahanty et al., 2017*). Significant commercialization of plant-growth-stimulating microbes and bio-control microbes is now occurring (*Kamilova et al., 2015*), and markets for microbial inoculants (mixtures of one or a few beneficial microbial organisms) are now growing faster than global markets for agrochemicals (*Batista & Singh, 2021*). Companies like BASF, DuPont, Bayer, Novozymes and Verdesian have become the top five companies for producing and selling microbial inoculants (*Sammauria et al., 2020*). However, a majority of the microbial inoculants are single to a small number of microbial species, synthetic microbial communities, novel prebiotics, and culture-dependent or independent microbes/microbiomes (*Batista & Singh, 2021*).

Microbial inoculants are expected to enhance microbial influence on plant growth, crop productivity, plant health and other associated agroecosystem functions and services, offering potential nature-based solutions to solve multiple climate issues (*Eggermont et al., 2018*). All members of the soil food web work collectively and simultaneously, as an assemblage of interacting processes (*Nicholson & Dupre, 2018*) operating in a web of relationships to form a fabric of coexistence: however, most studies, on soil microbiota have focused on a single species of microbes or on single attributes, viewing the microbial world from a reductionist perspective for evaluating microbial plant-growth-promoting characteristics (*Saharan & Nehra, 2011*).

Research by *Liu, Le Roux & Salles (2022)*, on the use of microbial inoculants to improve crop productivity and to assist in climate-change mitigation, observed many of the expected benefits demonstrated: low efficacy, inconsistent performance, or acted unpredictably in the field. These observations could result, when using inoculants that are incapable of assimilating into the resident soil microbiome, or are not beneficial in the form or function of their community diversity or structure.

*Edwards (1998)* and *Aguilar-Paredes et al. (2023)*, observed that vermicompost possesses outstanding biological properties and has microbial populations significantly larger and more diverse when compared to conventional composts. *Bhattacharyya & Jha (2012)* observed microbial communities in vermicompost promote: (a) the production of phytohormones (plant growth promoting hormones); (b) biological nitrogen fixation (both free-living and symbiotic); (c) phosphorus solubilization; (d) rhizosphere engineering; (e) quorum sensing; (f) biofilm formation; (g) humic matter production; (h) oxidation of metals from soil parent material for plant assimilation; (i) antibiotic production; (j) CO oxidation; and (k) the biodegradation of xenobiotics (pesticides, herbicides and plastics). *Pathma & Sakthivel (2012)* determined that vermicomposting amplifies the diversity and population of beneficial microbial communities for increasing soil fertility, enhancing plant growth and suppressing the population of pathogens and

pests. Research, on vermicomposts has also documented the presence of significant quantities of plant growth regulators such as auxins, cytokinin, and gibberellins, of microbial origin (*Krishnamoorthy & Vajranabhiah, 1986*; *Tomati, Grapppelli & Galli, 1988*; *Muscolo et al., 1993*) and humic acids (*Atiyeh et al., 2002*).

One issue that complicates experimentation in agricultural fields, is the limitation of understanding that these multicomponent agricultural "systems" are very complex and dynamic in their interrelated activities and are "difficult to comprehend" when implementing single discipline experimentation" (*Neal, Ritz & Crawford, 2020*). What has been missing, on these nature-based solutions, are large-scale "systems-oriented" field-scale applications and controlled field experiments to quantify the actual benefits of implementing microbial inoculations for restoring fertility and productivity in soils of agroecosystems.

This research employed a Biologically Enhanced Agricultural Management (BEAM) approach in a "systems-focused" research, on the adoption of regenerative practices, implementing: (a) no-till, (b) reductions in synthetic fertilizers, herbicides and insecticides, (c) continuous plant cover with "diverse species" winter cover and cotton commodity crops, and (d) the injection of an extract of vermicompost, sourced from a Johnson-Su bioreactor, into the furrow at each planting. The injection of this vermicompost extract is designed to provide a multi-species microbial community (archaea, bacteria, fungi, protozoa, algae, nematodes, microarthropods, viruses, *etc.*) to assist in developing the health and biological diversity of the plant/soil holobiome.

The experimental design, in this research project, adopted a 4-year time-series field approach, evaluating outcomes of applying the BEAM approach on a 5.2-hectare agricultural field, observing change-over-time in SOC% and TSN% and off-season cover crop biomass productivity. Baseline soil metrics were established in 2019, and after treatments, in years 2020, 2022 and 2023 to determine the change-over-time in SOC% and TSN%. Complimentary to this research, were three levels of synthetic N application, 0%, 15% and 100% of recommended application rates, focusing on synthetic nitrogen fertilizer's influence on: SOC%, TSN% increase/decrease. Other objectives of this research were to assess how the introduction of a BEAM "systems" approach, into a conventionally-managed cotton production system in Turkey influences reductions in: (a) herbicide, pesticide applications, (b) equipment and diesel use; and, increases in: (a) cover-crop biomass and cotton productivity, (b) economic return, and (c) field observable soil health improvements, over a 4-year research period.

This research hypothesizes that a BEAM "systems" approach, that employs full-time plant cover, by either commodity and/or annual off-season multi-species cover-crops, accompanied with applications of a microbially diverse vermicompost-derived extract from a Johnson-Su bioreactor, injected in-furrow at each planting, will promote: (a) enhanced SOC% and TSN% accumulation with more efficient capture and assimilation of atmospheric C ($CO_2$) as SOC; (b) improved agroecosystem soil fertility and productivity, and (c) increased farm profitability.

## MATERIALS AND METHODS

### Site location and seed selection

The project was a joint research effort, between Stella McCartney LTD (https://www.stellamccartney.com/us/en/), a luxury fashion designer, in collaboration with SÖKTAŞ (https://www.soktas.com.tr/), a specialist cotton producer and cotton-blended fabric manufacturer in Turkey, on farmland owned by SÖKTAŞ. The focus of this project was to identify and promote sustainable, regenerative agricultural management practices that could help achieve a 45% reduction in the C-footprint of the cotton fiber production portion of their manufacturing chain, towards satisfying the United Nations' net-zero emissions goal of reducing global greenhouse gas emissions 45% by 2030.

The research project began in the spring of 2019 on a 5.22-hectare field plot (Lat. 37.779322°, Lon. 27.494952°), 8.5 kilometers from Soke, Turkey (Fig. S1). The field was previously conventionally managed for many decades in a cotton/sunflower rotation with conventional tillage (rip, plow and disc) and bare fallows. Tillage operations were discontinued in fall of 2018 with no-till practices implemented from that point forward. The soil type, on the experimental plot, was a fluvi-calcaric fluvisol (FAO85; Type Jcf) with a sand (40%) silt (37%) clay (13%) texture. All field trials utilized local rainfall, or were flood irrigated (when necessary), during the summer commodity and winter cover-crop growing seasons.

The winter multi-species cover-crop seeds and the summer cotton seed, were planted with a no-till drill, outfitted with an injection system ensuring that the injected vermicompost extract (2.2 kg vermicompost in 187 L water ha$^{-1}$) comes in contact with the seed and soil as the seeds are planted. Winter cover-crops were multi-species mixes selected to have a majority percentage of legume seed varieties and are described in Table S1.

No fertilizers were used on the winter cover-crops and they were allowed to mature and reach early reproductive phase (approximately 50% plant blossom status, or early dough-stage for grain) and then rolled down and no-till planted into, with the spring-planted cotton commodity crops.

The cotton seed used in the trials in 2020 and 2021 was a local variety, "Meander 71", a unique long-staple hybrid cotton variety developed and grown by SÖKTAŞ. The 2022 season used Golden West's "Bomba" cotton seed. Both the "Meander 71" and "Bomba" varieties of cotton seed were planted at 100,000 seeds ha$^{-1}$ on 76.2 cm row spacing.

### Compost extract production

Vermicompost, from a Johnson-Su vermicomposting bioreactor, was utilized to produce the microbiological extracts that were injected into the furrow at each planting. The Johnson-Su bioreactor (https://youtu.be/DxUGk161Ly8) maintains an undisturbed, static (no turn) composting environment for a minimum of 1 year, 70% moisture content, aerobic conditions throughout the composting process, never allowed to freeze, with the addition of earthworms (*Eisenia fetida*) after the vermicompost bioreactor temperature decreases below 28 °C. The year-long composting process results in a vermicompost

having the texture of damp clay with no identifiable plant residues observable in the finished vermicompost matrix.

The microbiological extracts, made from this vermicompost, were produced in an extractor (https://www.youtube.com/watch?v=8ADdXIFdsqo) designed to separate the microbiota from any undecomposed plant residue. Water is continuously recycled in the extraction process and sprayed over the compost substrate in the extractor to produce a concentrated microbial extract that is then poured through a 200-mesh screen and then diluted and injected into the furrow at planting, to obtain liquid application rates of 187 liters ha$^{-1}$ containing ~2.2 kg vermicompost ha$^{-1}$.

## Cover crop and commodity crop trials

The winter cover-crop plant biomass, in this research, was rolled down with a roller/crimper just before planting the spring cotton crop and left on the surface to form a dense surface layer of biomass to suppress weed germination and help block sunlight on the soil surface to conserve water resources. No herbicides were used to terminate cover-crops. Two initial multispecies cover-crops (Table S1) were planted, at the beginning of the research, Spring (2019) and Fall (2019) on the test plots and then between each summer season's cotton commodity-crop. Three experimental fertilization-rate treatments were implemented in the cotton trials: (1) a BEAM +100% N approach, adopting the BEAM approach on cotton with a total annual application of ~203 kg elemental N ha$^{-1}$ (comprised of multiple applications of: 15-15-15; urea; ammonium nitrate; and ammonium sulfate); in addition to an injection of the microbial extract in-furrow at each planting, made with vermicompost, applied at a rate of 2.2 kg vermicompost in 187 liters water ha$^{-1}$; (2) a BEAM +15% N approach, applying 15% of the recommended inorganic nitrogen, (~30 kg elemental N ha$^{-1}$), in addition to the in-furrow injection of the microbial extract; and (3) a BEAM +0% N approach, using no application of synthetic N, and only the addition of the in-furrow injection of the microbial extract.

## Total soil organic carbon and total soil nitrogen analyses

Prior to initiating research trials, baseline soil samples were collected at six randomly chosen locations, at three depths 0–15, 15–30 and 30–45 cm across the 5.22 ha research plot. The soil samples consisted of a composite of three representative soil cores for each sampling location and profile depth, taken in April of each sampling year (2019, 2020, 2022, and 2023) at each of the six locations (Table S2). No soil sampling was conducted in 2021. Samples were labeled and placed in one-quart plastic bags and shipped to Selçuk University (Selçuk Üniversitesi, Ziraat Fakültesi, Aleaddin Keykubat Yerleşkesi Selçuklu-Konya-TURKEY) for SOC% and TSN% analysis. Laboratory analyses for SOC% assessment included a preliminary acidification treatment to remove soil carbonate content with follow-up dry-combustion using LECO protocols (*Wang & Anderson, 1998*). Total Soil Nitrogen was determined by an elemental analyzer through dry combustion.

Soil bulk density measurements were conducted for all three soil profile depths at each sampling location in each sample year of the research trial, using protocols adopted from the Food and Agriculture Organization's standard operating procedure for soil bulk

density (*FAO, 2023*) using soil core cylinders. Soil bulk density values were measured in g·cm$^{-3}$ and are available in Table S2.

## Commodity crop yield and economics

Cotton production was measured by harvesting and weighing each treatment's (SÖKTAŞ, BEAM +100% N, BEAM +15% N and BEAM +0% N) seed cotton production (seed and lint). The economic components included in the comparison between SÖKTAŞ's conventional cotton and the three BEAM-N% treatments for the years 2020, 2021 and 2022 are limited to items that reflect either a reduction in inputs or an increase in soil fertility related metrics all expressed in ($US ha$^{-1}$) including: (a) total seed cotton yield (t ha$^{-1}$) with cotton lint estimates assuming lint mass to be 35% of the seed cotton mass (*Goodman & Monks, 2003*); (b) gross revenue on the lint mass with cotton lint estimated at each production year's November market price for cotton lint (*Macrotrends, 2024*), (c) applied N fertilizer (nitrogen cost $2.59 kg$^{-1}$; (d) annual diesel fuel expenses for field operations (*Dobbins et al., 2015*; *U.S. Energy Information Administration, 2024*) (Table S3); (e) average cover-crop seed expenses; (f) chemical expenses for weed and insect control (SOKTAS); and (g) maintenance and operations costs (*Edwards, 2015*) (Table S4). Table 1 depicts each of these components and the difference in profitability between the three BEAM +N% treatments compared to SÖKTAŞ's conventional approach. Increases in TSN mass, to reflect the observed increase in soil nitrogen content over the 4-year research time frame, were estimated at $2.59 kg$^{-1}$ and limited to the crop-useable N, based on the SOKTAS conventional N application rates.

## Farm manager field observation

The farm managers in Turkey provided key field operations and observations: measuring pesticide application, amounts and type, farm equipment diesel use, irrigation amounts, crop performance, weed and insect pressures, cover-crop seed mixtures, cover-crop growth, aboveground cover-crop biomass metrics, and soil earthworm population dynamics.

## Statistical data analyses

Univariate statistical analyses were conducted to test differences in the means between pooled and paired 2019 baseline measurements, with follow-up 2020, 2022 and 2023 measurements, on the three BEAM N% treatments for: soil organic carbon percent (SOC %), Total soil nitrogen percent (TSN%), cover-crop biomass production (g dry biomass m$^{-2}$), seed cotton and cotton lint yield (kg ha$^{-1}$), soil bulk density (g cm$^{-3}$), operational costs (U.S. dollars), and reduction in inputs and increases in soil-fertility-related metrics. Statistical data analysis was conducted with Statistics Kingdom 2017, https://www.statskingdom.com/. All data components were initially examined for normal distribution with Shapiro-Wilk ($n < 50$); and F-tests were used to determine data variance. One baseline sample set was removed as an outlier for: (a) potential mis-labeling of a field sample collection, where SOC% values for 0–15 cm and 30–45 cm soil profile depth were reversed when compared to other observed mean SOC% measurements, (b) follow-up SOC% measurements, at that location, reversed to expected soil profile values in each of the

**Table 1 Change-over-time analyses of soil bulk density, SOC and TSN.**

| 0–15 cm | SOC% | BD (gm cm$^{-3}$) | SOC (t C ha$^{-1}$) | TSN (%) | N (t N ha$^{-1}$) |
|---|---|---|---|---|---|
| 2019 | 0.39 | 1.558 | 9.11 | 0.0317 | 0.741 |
| 2023 | 1.83 | 1.307 | 35.88 | 0.162 | 3.176 |
| | | 4-year change in SOC | 26.76 | 4-year change in TSN | 2.44 |
| | | Annual change in SOC | 6.69 | Annual change in TSN | 0.61 |
| **15–30 cm** | **SOC%** | **BD (gm cm$^{-3}$)** | **SOC (t C ha$^{-1}$)** | **TSN (%)** | **N (t N ha$^{-1}$)** |
| 2019 | 0.87 | 1.576 | 20.57 | 0.072 | 1.702 |
| 2023 | 0.064 | 1.53 | 14.69 | 0.062 | 1.423 |
| | | 4-year change in SOC | −5.88 | 4-year Change in TSN | −0.28 |
| | | Annual change in SOC | −1.47 | Annual Change in TSN | −0.07 |
| **30–45 cm** | **SOC%** | **BD (gm cm$^{-3}$)** | **SOC (t C ha$^{-1}$)** | **TSN (%)** | **N (t N ha$^{-1}$)** |
| 2019 | 0.51 | 1.496 | 11.44 | 0.038 | 0.853 |
| 2023 | 0.79 | 1.429 | 16.93 | 0.066 | 1.415 |
| | | 4-year change in SOC | 5.49 | 4-year change in TSN | 0.56 |
| | | Annual change in SOC | 1.37 | Annual change in TSN | 0.14 |
| | | SOC (t C ha$^{-1}$) 0–45 cm | 6.59 | TSN (t N ha$^{-1}$) 0–45 cm | 0.68 |

**Note:**
Baseline 2019 and 2023 soil organic carbon (SOC) and total soil nitrogen (TSN) mass for the 4-year research period and average annual change-over-time in the 0–15, 15–30 and 30–45 cm soil profiles. Abbreviations: Bulk density (BD); nitrogen (N).

following year's sampling, and (c) statistical testing regarded this location as an outlier. Means testing methodologies were selected based on the equality of variances and normality of data. Single factor ANOVA and Tukey HSD was implemented if multi-comparison data (three or more) were normally distributed with equal variance. If multi-comparison data was not normally distributed and/or variances were unequal, a Kruskal-Wallace means test was implemented with multiple comparisons, effect size, test power, outliers and a *post-hoc* Dunn's test. Pairwise comparisons implemented a two sample T-Test (pooled-variance) for known equal standard deviation and/or a two sample T-Test (Welch's T-test) for unknown unequal standard deviation data sets. Two-way analysis of variance (ANOVA), or factorial ANOVA—balanced design or, an unbalanced two-way ANOVA, factorial ANOVA—Unbalanced design were used to assess: fixed effects, mixed effects, random effects and mixed repeated measures of field data. Relative standard error (RSE) was used to measure and estimate data reliability using:

$$RSE = 100 \times [SE(r)/(r)] \qquad (1)$$

where SE(r) = the standard error of the estimate and (r) = the estimate. Results are reported as mean values, ± standard error (mean ± SE) and significance value α ≤ 0.05 threshold was used to determine statistical significance.

## RESULTS

### Field soil bulk density

Initial baseline field soil bulk density measurements relayed little difference of means, between soil profiles, 0–15 cm, μ = 1.558 ± 0.03 g cm$^{-3}$; 15–30 cm, μ = 1.57 ± 0.05 g/cm$^{3}$;

and 30–45 cm, $\mu = 1.50 \pm 0.06$ g/cm$^3$ (One-way ANOVA, F = 1.77; degrees of freedom (df) = (2,9), $p$-value = 0.23; Effect size = 0.63). Bulk density measurements, for the 2023 soil profiles were: $\mu = 1.31 \pm 0.04$ g cm$^{-3}$ for 0–15 cm; $1.53 \pm 0.02$ g cm$^{-3}$ for 15–30 cm and $1.43 \pm 0.02$ g cm$^{-3}$ for 30–45 cm. Relative standard error of each sampling period's bulk-density measurements indicated uniformity of means demonstrating less than 3.7% variance establishing the reliability for using baseline and 4-year soil bulk density values for SOC and TSN mass data analyses.

The 2023 soil bulk density in the 0–15 cm soil profile ($\mu = 1.31$ g cm$^{-3}$) was significantly different from baseline soil bulk density ($\mu = 1.56$ g cm$^{-3}$) (Two sample T-Test (Welch's T-test); T = 4.93; df = 12.99; $p = 0.0003$; Effect = 2.38). Final soil bulk density in the 15–30 cm soil profile ($\mu = 1.54$ g cm$^{-3}$) was significantly different from the baseline soil bulk density ($\mu = 1.62$ g cm$^{-3}$) (Mann-Whitney U test calculator; test statistic Z = −2.0679, $p = 0.04$; standardized effect size = 0.53). The 2023 soil bulk density in the 30–45 cm soil profile ($\mu = 1.43$ g cm$^{-3}$) was not significantly different from the baseline soil bulk density ($\mu = 1.55$ g cm$^{-3}$) (Mann-Whitney U test calculator; test statistic Z = −0.928, $p = 0.33$; standardized effect size = 0.24). The values for the mean soil bulk density measurements, for baseline and final soil profiles were used in the calculation of the increase or decrease in mass of SOC and TSN (t ha$^{-1}$ yr$^{-1}$).

## Soil organic carbon %

Analysis of the precision of the measured means, for the SOC% values of all three soil profiles (0–15 cm; 15–30 cm; and 30–45 cm) for the 4-year research period (2019, 2020, 2022, and 2023), demonstrated all SOC% values for all sample periods were normally distributed, and within the 95% confidence interval, when subjected to linear regression analyses. Each sample set, for 2019, 2020, 2022, and 2023 sampling events, exhibited a low-variance distribution and coefficients of variation for each sample set were less than one. Soil sample values, for SOC% and TSN%, were consistent across the field for baseline sampling in 2019 and as well for the follow-up soil sampling in 2020, 2022 and 2023. Relative standard error (%), for baseline 2019, and follow-up 2020, 2022 and 2023 SOC% and TSN% measurements, averaged less than 12.5% in the first year's sampling, decreasing to 10.1% in 2020, 9.6% in 2022 and 8.0% in 2023, values well within accepted RSE values (*Sileshi, 2015*) providing dataset reliability to accurately assess changes in SOC% and TSN%.

The 2019 sample average of SOC% in the 0–15 cm soil profile ($n = 5$) was $\bar{x} = 0.39 \pm 0.05$%; and the average of the 2023 SOC% in the 0–15 cm soil profile ($n = 9$) was $\bar{x} = 1.83 \pm 0.145$ SOC%; demonstrating a statistically significant increase of 1.44 SOC% between 2019 and 2023 (two sample T-Test (Welch's T-Test); T = −9.3719; df = 9.648; $p = 0.000004$; Effect = 3.97) (Fig. 1A). The SOC% for each year (2019, 2020, 2022 and 2023) increased linearly ($y = 0.3161x + 0.156$; $r^2 = 0.98$; std. dev ($S_{res}$) = 0.14871; slope $b_1 = 0.3161$; $p = 0.02$) at an average annual rate increase of 0.35% SOC yr$^{-1}$ over the 4-year research period in the 0–15 cm soil profile (Fig. 2).

The 2019 sample average of SOC% in the 15–30 cm soil ($n = 5$) profile was $\bar{x} = 0.87 \pm 0.06$ SOC%; and the 2023 SOC% in the 15–30 cm soil profile ($n = 9$) was $\bar{x} = 0.64 \pm 0.06$

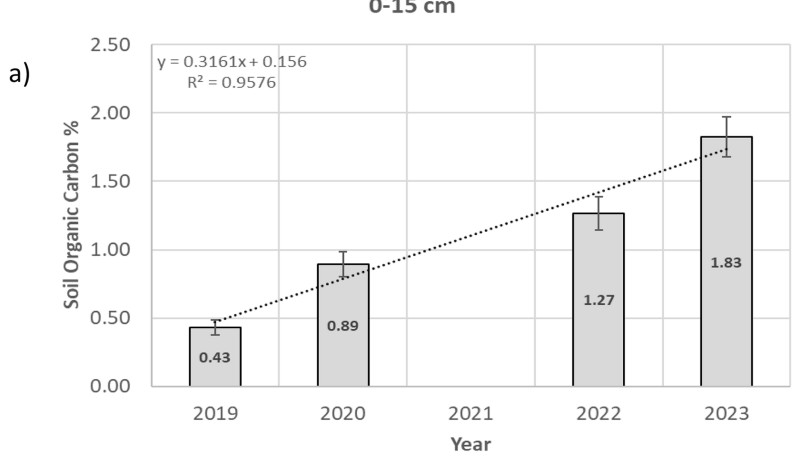

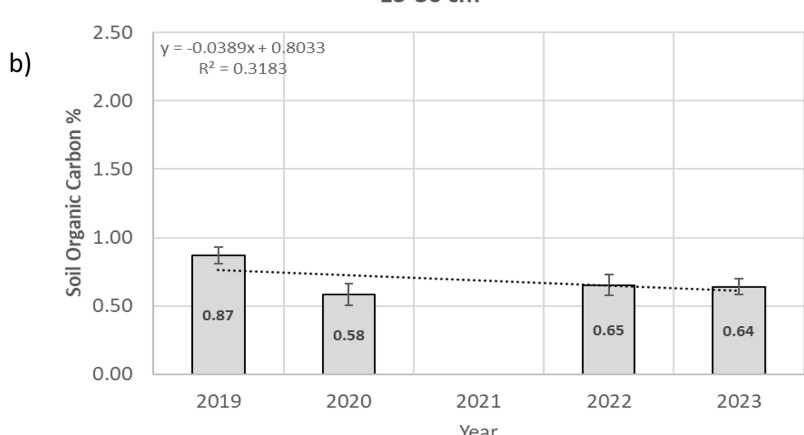

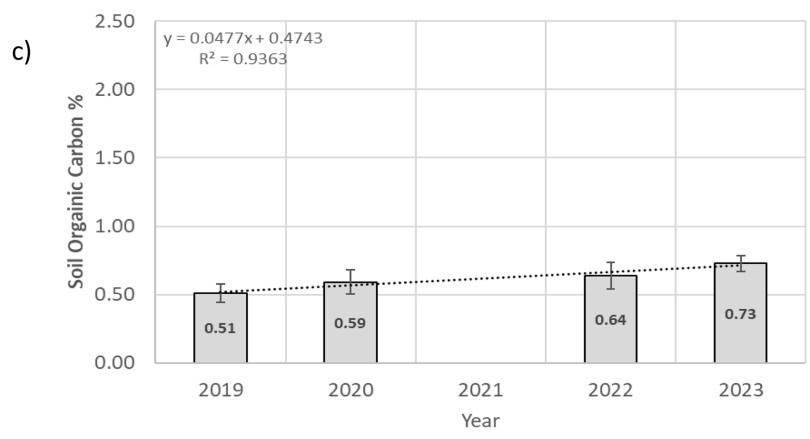

**Figure 1 (A–C) Soil organic carbon changes.** Soil organic carbon changes for years 2019, 2020, 2022, and 2023 for soil profiles: 0–15, 15–30 and 30–45 cm.

SOC% demonstrating a statistically significant loss of −0.23 SOC% between 2019 and 2023 (two-sample T-Test (Welch's T-Test); T = 2.6637; df = 10.0811; *p = 0.02*; effect = 1.39) (Fig. 1B).

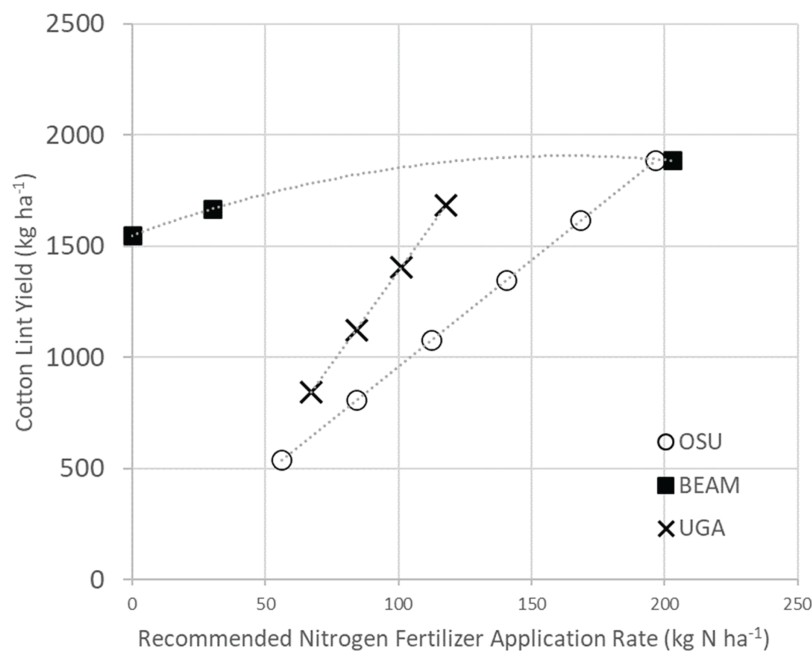

**Figure 2 Nitrogen fertilizer application rates.** Nitrogen fertilizer application rates (kg N ha$^{-1}$) for BEAM +15% N, compared to Oklahoma State University (OSU) and University of Georgia's (UGA) recommended nitrogen fertilization rates to achieve comparable cotton-lint yield (kg ha$^{-1}$).

The 2019 sample average of SOC% in the 30–45 cm soil profile ($n = 5$) was $\bar{x} = 0.51 \pm 0.07$ SOC%; and the 2023 SOC% in the 30–45 cm soil profile ($n = 9$) was $\bar{x} = 0.79 \pm 0.08$ SOC% demonstrating a statistically significant increase of 0.28 SOC% between 2019 and 2023 (Two-sample T-Test (Welch's T-Test); T = −2.6646; df = 11.631; $p = 0.02$; effect = 1.3) (Fig. 1C).

Pooled field results, for change-over-time SOC% measurements, were calculated, subtracting the mean of 2019 SOC% from the mean of the 2023 SOC%, to assess the increase or decrease in SOC% in each of the soil profiles 0–15, 15–30 and 30-45 cm over the 4-year study period. The total SOC mass, over the 4-year research period, increased by 26.37 metric tons (t) SOC ha$^{-1}$ in the 0–45 cm soil profile (Table 1).

The annual increase in SOC mass in the 0–15 cm soil profile averaged 6.69 t SOC ha$^{-1}$ yr$^{-1}$. The 15–30 cm soil profile SOC mass decreased by 1.47 t SOC ha$^{-1}$ yr$^{-1}$, and the 30–45 cm soil profile SOC mass increased by 1.37 t SOC ha$^{-1}$ yr$^{-1}$ for a cumulative average annual increase of 6.59 t SOC ha$^{-1}$ yr$^{-1}$ when considering the annual change in SOC for all three soil profiles, in the 0–45 cm soil profile, over the 4-year study period (Table 1).

### Total soil nitrogen%

The 2019 sample average of TSN% in the 0–15 cm soil profile ($n = 5$) was $\bar{x} = 0.032 \pm 0.005$ TSN% and the 2023 TSN% in the 0–15 cm soil profile ($n = 9$) was $\bar{x} = 0.162 \pm 0.014$ TSN% demonstrating a statistically significant increase of 0.131 TSN% between 2019 and 2023 (two-sample T-Test (Welch's T-Test); T = −8.6362; df = 6.0946; $p = 0.0001$; effect = 4.9) (Table 1).

The 2019 sample average of TSN% in the 15–30 cm soil profile ($n = 5$) was $\bar{x} = 0.072 \pm 0.006$ TSN%; and the 2023 TSN%, in the 15–30 cm soil profile ($n = 9$), was $\bar{x} = 0.062 \pm 0.005$ TSN% demonstrating a statistically non-significant decrease of −0.024 TSN% between 2019 and 2023 (two-sample T-Test (Welch's T-test); T = 1.3813; df = 8.3031; $p = 0.2$; effect = 0.84) (Table 1).

The 2019 sample average of TSN% in the 30–45 cm soil profile ($n = 5$) was $\bar{x} = 0.038 \pm 0.006$ TSN%; and the 2023 TSN% in the 30–45 cm soil profile ($n = 9$) was $\bar{x} = 0.066 \pm 0.007$ TSN% demonstrating a statistically significant increase of 0.02 TSN% between 2019 and 2023 (two-sample T-Test (Welch's T-test); T = −2.443; df = 7.8343; $p = 0.04$; effect = 1.4) (Table 1).

Cumulative TSN mass increased 2.72 t TSN ha$^{-1}$ in the top 45 cm of the BEAM soil profiles, over the 4-year period of this research, at an average annual rate of 0.68 t TSN ha$^{-1}$ yr$^{-1}$ (Table 1).

There was no statistical difference in the increase of TSN% values between the 100% +N, 15% +N and 0% +N treatments in the 0–15 cm soil profile (One-way ANOVA test, using F distribution, df (2,6) (right tailed); f = 0.15; F = 0.07; $p = 0.94$). There was no statistical difference in the decrease of TSN% values between the 100% +N, 15% +N and 0% +N treatments in the 15–30 cm soil profile (one-way ANOVA test, using F distribution, df (2,6) (right tailed); f = 0.71; F = 01.51; $p = 0.29$). There was no statistical difference in the increase of TSN% values between the 100% +N, 15% +N and 0% +N treatments in the 30–45 cm soil profile (one-way ANOVA test, using F distribution, df (2,6) (right tailed); f = 0.65; F = 01.25; $p = 0.35$).

The individual measurements of TSN% were linearly correlated with the corresponding SOC% measurements ($r^2 = 0.99$), indicating that TSN was closely associated or contained within the SOC structure, or organic matter, maintaining a relatively constant C:N ratio of 11.5:1. Raw data for individual and cumulative soil organic carbon (SOC%) and total soil nitrogen (TSN%) increases/losses, as compared to soil depth (0–15, 15–30 and 30–45 cm) over the 4-year research period, are reported in Table S2.

## Commodity (Cotton) crop yields and profitability

Annual cotton crop yields for each year's (2020, 2021, 2022) BEAM "regenerative" cotton/cover-crop rotations for the BEAM +100% N (203.5 kg (N) ha$^{-1}$) were: (a) 4.72, 4.16, and 5.26 t seed cotton ha$^{-1}$ (lint 1.652, 1.456, and 1.841 t lint ha$^{-1}$). Cotton crop yields for 2020, 2021, and 2022 BEAM +15% N, (30.5 kg N ha$^{-1}$) were 4.06, 4.21 and 4.24 t seed cotton ha$^{-1}$ (lint production 1.421, 1.474 and 1.484 t lint ha$^{-1}$). Cotton crop yields for 2020, 2021 and 2022 for the BEAM +0% N were 3.82, 3.85 and 3.94 t seed cotton ha$^{-1}$ (lint production 1.337, 1.348 and 1.375 t lint ha$^{-1}$) (Table 2).

The 2020 BEAM treatments (100% N, 15% N, and 0% N) demonstrated $290.97, $360.54 and $310.12 ha$^{-1}$ net profit increase/(decrease) respectively, when compared to SÖKTAŞ (2020) conventional profit margins (Table 2). The 2021 BEAM treatments (100% N, 15% and 0% N) demonstrated $(−220.33) ha$^{-1}$, $209.56, and $98.24 ha$^{-1}$ respectively net profit increase/(decrease) when compared to SÖKTAŞ (2021) conventional profit margins (Table 2). The 2022 BEAM treatments (100% N, 15% and 0% N)) demonstrated

**Table 2  Total revenue itemization for 2020, 2021, and 2022.**

| 2020 | SOKTAS average (4.61 tons/ha) | BEAM +100% N (4.72 tons/ha) | BEAM +15% N (4.06 tons/ha) | BEAM +0% N (3.82 tones/ha) |
|---|---|---|---|---|
| Lint (tons)/ha | 1.6135 | 1.652 | 1.421 | 1.337 |
| Gross revenue | $2,339.58 | $2,395.40 | $2,060.45 | $1,938.65 |
| Fertilizer | $(475.90) | $(475.90) | $(71.38) | $- |
| Diesel for operations | $(83.99) | $(26.75) | $(26.75) | $(26.75) |
| Labor/Maintenance | $(139.22) | $(69.61) | $(69.61) | $(69.61) |
| Cover crop seed cost ($/ha) | $- | $(111.00) | $(111.00) | $(111.00) |
| Herbicide ($/ac) | $(219.30) | $- | $- | $- |
| Pesticide ($/ha) | $(81.78) | $(81.78) | $(81.78) | $(81.78) |
| **Net profit** | **$1,339.39** | **$1,630.36** | **$1,699.93** | **$1,649.51** |
| Delta | BEAM *vs* SOKTAS | $290.97 | $360.54 | $310.12 |
| Percent difference | Profit Increase % | 22% | 27% | 23% |
| **2021** | **SOKTAS average (5.075 tons/ha)** | **BEAM +100% N (4.16 tons/ha)** | **BEAM +15% N (4.21 tons/ha)** | **BEAM +0% N (3.85 tons/ha)** |
| Lint (tons/ha) | 1.77625 | 1.456 | 1.4735 | 1.3475 |
| Gross revenue | $2,566.68 | $2,111.20 | $2,136.58 | $1,953.88 |
| Fertilizer | $(475.90) | $(475.90) | $(71.38) | $- |
| Diesel for operations | $(83.99) | $(26.75) | $(26.75) | $(26.75) |
| Labor/Maintenance | $(139.22) | $(69.61) | $(69.61) | $(69.61) |
| Cover crop seed cost ($/ha) | $- | $(111.00) | $(111.00) | $(111.00) |
| Herbicide ($/ac) | $(219.30) | $- | $- | $- |
| Pesticide ($/ha) | $(81.78) | $(81.78) | $(81.78) | $(81.78) |
| **Net profit** | **$1,566.49** | **$1,346.16** | **$1,776.05** | **$1,664.74** |
| Delta | BEAM *vs* SOKTAS | $(220.33) | $209.56 | $98.24 |
| Percent difference | Profit Increase % | -14% | 13% | 6% |
| **2022** | **SOKTAS average (5.23 tons/ha)** | **BEAM +100% (5.26 tons/ha)** | **BEAM +15% N (4.24 tons/ha)** | **BEAM +0% N (3.94 tons/ha)** |
| Lint (tons)/ha | 1.8305 | 1.841 | 1.484 | 1.379 |
| Gross revenue | $2,654.23 | $2,282.84 | $2,151.80 | $1,999.55 |
| Fertilizer | $(475.90) | $(475.90) | $(71.38) | – |
| Diesel for operations | $(83.99) | $(26.75) | $(26.75) | $(26.75) |
| Labor/Maintenance | $(139.22) | $(69.61) | $(69.61) | $(69.61) |
| Cover crop seed cost ($/ha) | $- | $(111.00) | $(111.00) | $(111.00) |
| Herbicide ($/ac) | $(219.30) | $- | $- | $- |
| Pesticide ($/ha) | $(81.78) | $(81.78) | $(81.78) | $(81.78) |
| **Net profit** | **$1,654.04** | **$1,517.80** | **$1,791.28** | **$1,710.41** |
| Delta | BEAM *vs* SOKTAS | $(136.24) | $137.24 | $56.37 |
| Percent difference | Profit Increase % | −8% | 8% | 3% |
| | SOKTAS | BEAM +100% N | BEAM + 15% N | BEAM + 0% N |
| Total revenue (2020–2022) (minus inputs) | **$4,559.92** | **$6,159.06** | **$5,267.25** | **$5,024.66** |
| Total 3 yr profit increase (SOKTAS *vs*. BEAM) | | **$(65.60)** | **$707.33** | **$464.74** |

| Table 2 (continued) | | | | |
|---|---|---|---|---|
| | SOKTAS | BEAM +100% N | BEAM + 15% N | BEAM + 0% N |
| Average profit increase (Loss) yr$^{-1}$ | | $(21.87) | $235.78 | $154.91 |
| Soil N increase ha$^{-1}$ y$^{-1}$ (590 kg ha$^{-1}$) total market value | | $1,378.77 | $1,378.77 | $1,378.77 |
| Useable soil N ha$^{-1}$ yr$^{-1}$ | | $475.90 | $475.90 | $475.90 |
| Nature-based carbon offsets ($8.86 ton$^{-1}$ @6.6 t C ha$^{-1}$) | | $214.37 | $214.34 | $214.37 |
| Total increase in revenue BEAM vs SOKTAS ($ha$^{-1}$) | | **$668.40** | **$926.02** | **$845.18** |

Note:
Annual (2020, 2021, and 2023) cotton production yields and net profitability by year relaying: lint mass, gross revenue, operational cost, net profit plus assessment of current (2024) market values for total soil nitrogen and soil organic carbon increases. Fuel costs are estimated in Table S3, and herbicide and pesticide costs are estimated in Table S4.

$(−136.24), $137.24 and $56.37 ha$^{-1}$ net profit increase/(decrease) respectively, when compared to SÖKTAŞ (2022) conventional profit margins (Table 2). The BEAM +15% N performed the best of all three BEAM treatments, over the 3 years of cotton production, promoting an average annual $235.78 ha$^{-1}$ yr$^{-1}$ increase in profitability when compared to traditional SÖKTAŞ management practices (Table 2). Profit gains and/or losses were a result of cotton lint yield and associated reduction in input costs (fertilizer, herbicides, insecticides, diesel and reduced labor/maintenance) enumerated in Table 2.

Field reservoir TSN% increased an average of ~680 kg N ha$^{-1}$ yr$^{-1}$, across all three BEAM treatments. The TSN assimilation cost-offset benefits of ~$1,570.80 ha$^{-1}$ yr$^{-1}$ would be realized, using the N fertilizer cost $ US ha$^{-1}$, $2.31 US kg$^{-1}$. The economic value of N (209 kg ha$^{-1}$), implemented in the SOKTAS approach would be ~$475.90 ha$^{-1}$. This value was used to assess the value of N made available to future cotton crops, towards offsetting synthetic N fertilizer applications based on average SOKTAS conventional N usage.

When considering "nature-based" carbon assimilation offsets of $8.86 ton$^{-1}$ $CO_2$ ($32.28 t$^{-1}$ C) (Ecosystem Marketplace, 2024) reimbursement for the 6.6 t C offsets would total $214.37 ha$^{-1}$ yr$^{-1}$. These increases for TSN and SOC would increase total annual increases in net profitability, for the BEAM +100% N; BEAM+15% N and BEAM+0%, an average of $668.40, $926.02 and $845.18 yr$^{-1}$, respectively (Table 2).

The BEAM +15% N approach, using 30.45 kg N ha$^{-1}$, averaging 1,458 kg cotton lint ha$^{-1}$, produced the most cotton with the least mass of synthetic N inputs in the BEAM trials. This cotton yield was compared with Oklahoma State University (OSU) (Arnal & Boman, 2017) and University of Georgia's (UGA) recommended nitrogen application rates, related to the expected mass of lint production per mass of nitrogen addition (kg ha$^{-1}$) (Shirley, 2021). The recommended mass of nitrogen necessary, to achieve the observed mass of cotton lint production in the BEAM +15% N, would have required an extra 95 kg N ha$^{-1}$ up to 156 kg N ha$^{-1}$ as determined by OSU and UGA respectively (Fig. 3).

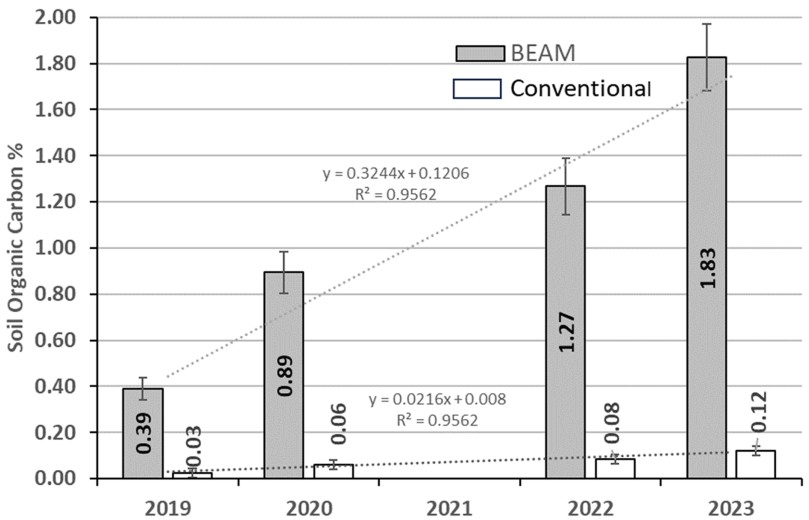

**Figure 3 Comparison of SOC% in BEAM and multiple agroecosystems.** Comparison of annual increases in BEAM SOC% (gray filled bars) to the average of several meta-analyses of SOC% increases in multiple agroecosystems (no-fill bars).

## Influence of fertilization rates and soil depth on SOC% and TSN %

A two-sample ANOVA statistical analysis was conducted to determine if the three fertilizer treatments (A: 100% N, 15% N and 0% N) administered for summer cotton production, influenced changes in 2023 SOC% at each of the three depths (B: 0–15, 15–30 and 30–45 cm). The fertilizer nitrogen application rates ($F_A$) demonstrated no statistically-significant influence, on the change in SOC% (Two-sample ANOVA—fixed-test, using F distribution (right-tailed); $F_A = 0.102$; *p = 0.9*; $n^2 = 0.011$). While no statistically significant difference was observed in the SOC%, when compared to the cotton N-fertilization rates, there was an observable trend developing where the 0% N treatment consistently had the greatest increase in SOC% in the 0–15 and 30–45 cm soil profiles, and the least loss of SOC% in the 15–30 cm soil profile (Fig. S2), when compared to the 15% N and 100% N treatments.

A trend, similar to what was observed with the fertilization rate and SOC% increases was observed for TSN%, with the BEAM +0% N and BEAM +15% N demonstrating the highest average increases in TSN%, while the BEAM + 100% N experienced the lowest TSN% increase, even when considering the annual applications of 203.5 kg elemental N% ha$^{-1}$ in each of the three annual commodity crop plantings (Fig. S3).

## Farm management field observations

The farm managers observed: (a) increases in earthworm populations, increasing from zero earthworms m$^{-2}$ observed in the 2019 soil profile, to ~100 earthworms m$^{-2}$ in 2023 (average of ten field sampling points); (b) annual cover crop productivity increased from ~400 g dry biomass m$^{-2}$ in 2020–2021, ~682 g dry biomass m$^{-2}$ in 2021–2022 to ~925 g dry biomass m$^{-2}$ in 2022–2023 (average of three field sampling points) without the application of synthetic fertilizers; and (c) the three different cotton commodity-crop N-fertilizer application rates appeared to have no influence on the uniformity of winter cover

crop growth between the 100%, 15% and 0% N treatments and the following year's sample values for TSN%.

The farm managers also tracked herbicide and pesticide application rates, diesel and irrigation water use in the BEAM treated plots, observing: (a) 100% reduction in herbicide application, (b) 56% reduction in insecticide application (c) 65% reduction in tractor diesel use, and farm labor, due to fewer trips across the field, and (d) the potential to reduce irrigation water ~30% (due to the rolled down winter cover crops for soil surface protection; and cotton yields being maintained during a water shortage in Turkey with government mandates for reduction in the number of irrigation events from three to two).

A local farmer that has been observing these BEAM managed fields also acknowledged: *"The greatest reward has been to witness the difference in soil health. Soil aggregation has been steadily increasing, and while a variety of tests measuring carbon and nitrogen in the soil and on the crop confirm this, the structural improvements are so apparent that they are visible to the naked eye."* (D. Johnson, 2024, personal correspondence with I. Uysal).

## DISCUSSION

A Biologically Enhanced Agricultural Management protocol, or BEAM "systems" approach was implemented, in this research project, implementing: (a) no-till practices, (b) no, or reduced synthetic fertilizer applications, (c) full-time plant cover, either commodity and/or annual multi-species cover-crops, accompanied with (d) application of a microbial vermicompost extract, from a Johnson-Su bioreactor, injected in-furrow at plantings. Soil fertility and productivity is dependent on the composition and activity of soil microorganisms for affecting soil structure, nutrient cycling, and organic matter decomposition (*Toor et al., 2024*). Addition of vermicompost, in other research efforts, has promoted multiple benefits for improving soil microbial community structure and related microbiological properties towards improving plant growth, as it contains: (a) multiple beneficial organisms: bacteria, fungi, actinomycetes that improve microbial diversity and activity (*Domínguez, Aira & Gomez-Brando, 2010*); (b) plant growth promoting rhizobacteria and mycorrhizal fungi that enhance plant nutrient uptake, and microbiota that promote improved microbial functional diversity, and (c) microorganisms that promote increases in soil enzyme activity (*Saini & Sharma, 2019*).

### System net primary productivity improvement

The implementation of cover-crops, in this research, was designed to optimize the net primary productivity for the 6 months that a field would normally lay fallow in a conventionally-managed cotton operation, allowing year-round plant cover and roots in the ground, as either a commodity crop or a multi-species cover-crop. The farm managers observed annual increases in cover-crop net primary productivity (NPP) of: ~400 g dry biomass m$^{-2}$ in 2021, ~682 g dry biomass m$^{-2}$ in 2022 and ~925 g dry biomass m$^{-2}$ in 2023, and they also noted that the cover crop biomass was evenly distributed across the field, regardless of summer commodity crop treatment N% application. In comparison: (a) *Jackson & Jackson (2000)* estimated annual NPP for cultivated land at ~650 g dry biomass m$^{-2}$; and (b) *Whittaker (1975)* observed the most productive terrestrial ecosystems, the

tropical rain forests, produce ~2,200 g dry biomass m$^{-2}$. The increase in NPP, in the 2023 plant cover-crop biomass, potentially represents ~9.25 tons of aboveground dry biomass ha$^{-1}$, yielding ~4.12 tons of C ha$^{-1}$ yr$^{-1}$ (~44.53% C mass ratio (*Acharya et al., 2015*)).

Aboveground biomass is not commonly assessed in many studies researching SOC accrual, but the Food and Agricultural Organization (*Ponce-Hernandez, Koohafkan & Antoine, 2004*) recommends considering both SOC, as well as estimates of C, in the aboveground biomass in "agriculture, pastures, crop rotations and mixes of crops". Supporting FAO's position, is research by *Prescott & Vesterdal (2021)*, where they observed the C in leaf litter, laying on the surface of the ground, degrades slowly, with estimates of ~43% of litter-C remaining after the first year, ~30% after the second year, ~23% by year three, eventually degrading to a constant of ~20% of the original litter-C mass remaining on the soil surface, through the 12 years their research was conducted. *Buckeridge et al. (2020)* observed that microbial transformations play key roles in the stabilization of surface plant litter C and that a large percentage of stable C is contained in stable "microbial necromass and transformation products".

The 2023 BEAM +15% N total cotton crop NPP was ~1,108 g m$^{-2}$ with cotton crop stalk biomass estimates based on seed cotton production (stalk biomass = 55% and seed cotton production = 45% (*Wanjura et al., 2014*)). The ~1,108 g m$^{-2}$ NPP of the 2022 cotton crop plus the 2023 cover-crop biomass of ~925 g dry biomass m$^{-2}$ almost doubles annual NPP in the BEAM "systems" approach, providing a total NPP of ~2,033 g dry biomass m$^{-2}$ yr$^{-1}$, an amount approaching ~92% of the NPP of tropical forests (2,200 g dry biomass m$^{-2}$). This cover-crop biomass, produced over the winter and early spring, provided: (a) ~4.12 tons cover-crop C mass; (b) soil-surface plant residues to help retain soil moisture, slow erosion and reduce soil surface temperature; (c) nutrients, and energy in the C-bonds and elemental nutrients in the cover-crop biomass to help sustain, and/or increase soil microbial community population, structure, diversity and biological functionality, and (d) nutrients for the next crop to assimilate, as the biomass, and the nutrients it contains, are made available for reuse by the soil microbiome (bacteria, fungi, *etc.*) and microarthropods and earthworms.

## Soil organic carbon increase

Many researchers have explored the potential to increase SOC through adoption of regenerative practices either singularly or in tandem including: no-till, reductions in fertilizers, cover crops, additions of biomass or compost and some have included vermicompost products. Several research efforts, on the use of vermicompost for increasing SOC, have consisted of: (a) small-scale greenhouse studies with vermicompost application rates ranging from 400 kg ha$^{-1}$ to 4 t ha$^{-1}$, applying vermicompost that had an incubation duration of 50 days (*Oroka, 2015*), or; (b) a 3-year mesocosm scale (1 m × 1 m × 2 m deep) with vermicompost application rates of 2 t ha$^{-1}$, while including synthetic chemical nutrients amendments, and a 90 day incubation period for the vermicompost product (*Ngo et al., 2014*); or (c) the applications rates of 5 to 10 t ha$^{-1}$ vermicompost based solely on nutrient values (*Abad & Shafiqi, 2024*). These vermicompost application rates are impractical for large-scale agriculture, and the short

duration of vermicompost incubation (from 50 to 90 days) has been observed to promote phytotoxicity and inhibit the potential for promoting plant growth (*Majlessi et al., 2012*).

This research applied vermicompost, from a Johnson-Su bioreactor, with a minimum of 1-year processing, at a practical field application rate of 2.2 kg ha$^{-1}$, viewing the vermicompost extract as a soil microbial community inoculant, providing little nutrient value for promoting crop productivity. This BEAM, fully regenerative "systems" approach, utilizing all listed regenerative practices along with injection of a Johnson-Su vermicompost inocula, designed to improve soil microbial community (SMC) diversity and functionality, displayed a consistent year-over-year annual average increase of ~6.59 metric tons C ha$^{-1}$ yr$^{-1}$ in the top 45 cm of the soil profile. The majority of the SOC increase was located in the top 15 cm of the soil profile (Table 1) increasing SOC% from 0.39% to 1.83%, over the 4-year research period, increasing at a rate of 0.36% yr$^{-1}$. Accompanying this increase in SOC mass was the total soil nitrogen mass increasing at a rate of ~0.68 t N ha$^{-1}$ yr$^{-1}$, following closely with the increases in SOC%, (X predicts Y, $R^2 = 0.99$, $F_{(1,25)} = 7,889.78$, $p < 0.001$). TSN% accrual rates, in the 0–15 cm soil profile, increased from 0.032% to 0.162%, over the 4-year period, at a rate of 0.032% yr$^{-1}$ (Table 1).

## Soil organic carbon stabilization

The increase in earthworm populations (from 0 to ~100 earthworms m$^{-2}$), observed by the farm managers, aligns with the measured improvements in SOC and TSN in this research, potentially providing a pathway for much of the above-ground surface biomass to transit through the earthworm community to produce surface and subsurface worm casts. These casts are known to contain significant quantities of highly-stable macro and micro-aggregates (*Kumar et al., 2023*), particulate organic matter (POM), and mineral associated organic matter (MAOM) (*Angst et al., 2021*). *Vidal et al. (2019)* observed that over 85% of the total organic carbon in worm casts was in the MAOM configuration "providing stable organo-mineral associations sequestering litter-derived C on longer timescales". *Angst et al. (2022)* determined, beyond bioturbation activity, earthworms "accelerate the formation of microbial necromass, stabilized in aggregates and organo-mineral associations of POM, and MAOM".

The mean population of earthworms in Fluvisol soils has been estimated at ~35.7 earthworms m$^{-2}$ in a global earthworm enumeration study conducted by *Phillips et al. (2021)*. Earthworm counts, in the cotton-research test plots in Turkey, increased from zero earthworms m$^{-2}$ in 2019 to ~100 earthworms m$^{-2}$ in 2023, approximately three times the expected mean population (*Phillips et al., 2021*) observed, as the soil fertility improved in the BEAM field treatments.

Earthworm cast can build soil profiles at ~0.2–0.56 cm yr$^{-1}$, and the mean amount of soil brought to the surface by the earthworms is estimated at ~17–40 t ha$^{-1}$ year$^{-1}$ (*Feller et al., 2003*). Earthworm casts can contain a range of carbon percentages, from 4.33–7.50% C (*Angst et al., 2022*) to an average of 21.3% C (*Chen et al., 2021*). Bulk density of vermicompost ranges between 0.854 to 0.658 g cm$^{-3}$ (*Bahram et al., 2014*). Based on these studies, the soil organic carbon in surface-deposited vermicast, could range between 0.48 to 7.3 t C ha$^{-1}$ yr$^{-1}$. These annual rates of surface vermicast deposition would assist in
rebuilding soil profiles, requiring from 26 to 75 years to re-establish a 15 cm deep topsoil profile as opposed to the median historical soil erosion loss rates of ~0.018 ± 0.012 cm year$^{-1}$, currently occurring in agricultural landscapes (*Thaler et al., 2022*).

Compatibility of microbial inocula with the existing soil microbiome is critical for successful assimilation. The microbiome structures in a vermicompost likely have a community structure that would easily assimilate into a soil microbiome, since earthworms have historically been instrumental in improving soils. Earthworm's burrowing and feeding activity has also been observed to have beneficial impacts on productivity (*Fonte, Hsieh & Mueller, 2023*). Adoption of the BEAM approach, in regenerative systems, appears to promote conditions that assist in the restoration of earthworm populations, helping to decompose cover crop biomass and promote deposition of vermicast for rebuilding carbon-rich soil profiles. These observations challenge the current perspectives that there is a limit to the amount of carbon that can be stored in agricultural soils, as soils are rebuilt and soil profile depths increase year-over-year.

### Original research goals

The original objectives of this research were to reduce SÖKTAŞ's carbon-footprint of cotton fiber production by 45%, towards accommodating current United Nations carbon reduction mandates, and evaluate increase in profitability when adopting a BEAM agricultural approach. Estimates for cotton's carbon-footprint (C) t$^{-1}$ lint, for total cotton production, (fiber, yarn, fabric and waste) was researched and determined by SÖKTAŞ for their operation in Turkey. Estimates for SÖKTAŞ's annual C footprint, for each hectare in production, (adjusted for the average 1.755 t ha$^{-1}$ of annual cotton lint production) were: (a) fiber production (0.42 t C); (b) yarn production (3.56 t C); (c) fabric production (0.357 t C) and (d) waste treatment (0.057 t C). All components totaled ~4.40 t C ha$^{-1}$ for SÖKTAŞ's complete annual C footprint of cotton fiber-to-fabric production (Fig. 4). The 6.59 t C ha$^{-1}$ yr$^{-1}$ increase in SOC, observed in the BEAM approach in this research, offset all the 0.42 t C for SÖKTAŞ's fiber production, surpassing the United Nations mandate for a 45% reduction of their fiber production, and additionally, it offset the C footprint of their annual yarn production, fabric production and treatment of waste with 2.19 t C ha$^{-1}$ remaining in reserve.

Looking at the complete BEAM cotton management system, and developing C-offset estimates to include farm manager observations for cover crop biomass production and worm population increases, the cumulative system-C assimilated, based on results of the fourth year of the BEAM approach, would be comprised of: (a) ~6.59 t C ha$^{-1}$ yr$^{-1}$ for SOC increase; (b) ~4.12 t of C ha$^{-1}$ yr$^{-1}$ for the C in the repeated annual planting and growth of cover-crops; (c) ~0.82 t C ha$^{-1}$ yr$^{-1}$ for residual cover-crop C in the 20% previous year's cover-crop biomass C, building up as necromass in the field from each previous years cover-crop growth (*Prescott & Vesterdal, 2021*); and d) ~0.77 t C ha$^{-1}$ yr$^{-1}$ for the C in the cotton lint, exported from the field and incorporated into fabric (*Hedayati et al., 2019*).

The C-mass that could be avoided, in the fourth year of this BEAM systems-approach, would be comprised of: (a) ~0.64 t C ha$^{-1}$ yr$^{-1}$ for adopting zero-till practices avoiding an

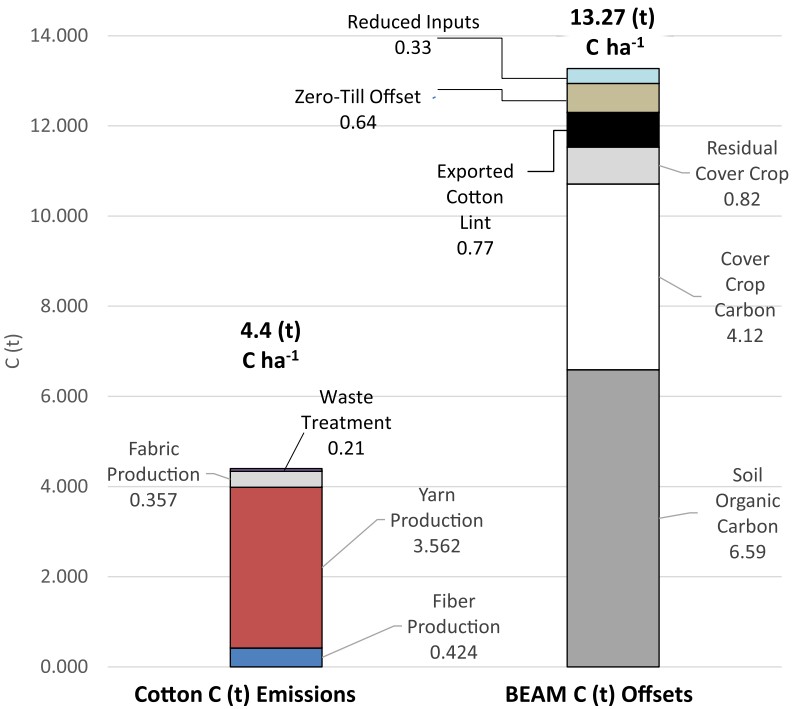

**Figure 4 Comparison of carbon (C) emissions for the production of fiber, yarn, fabric, and waste.**
Comparison of carbon (C) emissions for the production of fiber, yarn, fabric, and waste treatment for cotton production (SOKTAS), *vs* available carbon offsets in a biologically enhanced agricultural management (BEAM) protocol for soil organic carbon, cover crop carbon, cover crop residuals, exported cotton carbon, zero till offsets and reduced inputs (fuel, nitrogen phosphorus, herbicides and pesticides).

increase in $CO_2$ respiration emissions related to conventionally-managed soil disturbance (*Abdalla et al., 2016*); and b) ~0.33 t C ha$^{-1}$ yr$^{-1}$ for reduced inputs (nitrogen fertilizers, herbicide, insecticide and diesel) (*Hoxha & Christensen, 2018*; *Lal, 2004a, 2004b*) (Fig. 4).

The total carbon offset and avoided for this research, in a BEAM systems analysis, would total ~13.27 t C ha$^{-1}$ yr$^{-1}$. In this scenario, the 45% C reduction goal of SÖKTAŞ's fiber production C mass (~0.42 t C ha$^{-1}$ yr$^{-1}$) would be offset as well as the remainder of the 3.98 t C ha$^{-1}$ yr$^{-1}$ for yarn production, fabric production and treatment of process waste, while still providing 8.87 t C ha$^{-1}$ yr$^{-1}$ towards productive utilization of C, to offset other carbon emissions (Fig. 4).

The BEAM process is simple and practical from an implementation perspective and comes at no extra cost to the consumer or our society, as it can become part of a "business-as-usual" farm management approach. The environmental benefits expand out to other constituents of our society for land, water and air restoration.

All information, on the BEAM process and Johnson-Su bioreactor construction, is open-source, and freely available. Any size farming operation can easily produce the vermicompost necessary for the inoculant. One Johnson-Su bioreactor produces ~317 kg finished compost, capable of treating 144 hectares (350 acres). Application can be accomplished on large land areas with existing equipment (liquid fertilizer injections

systems already on planters or retrofitted no-till drills) in a one-pass planting/inoculation process. Cost for compost production and application is ~$2.40 to $3.70 ha$^{-1}$ and many of the bioreactor construction and composting materials are already available on the farm.

NOTE: Carbon assimilation/avoidance values were not estimated for the N mass accumulation observed in the soil profile. Soil carbon respiration, between BEAM, and SÖKTAŞ conventional approaches, were considered equivalent based on research conducted by *Breil et al. (2023)*.

## CONCLUSIONS

The adoption of a BEAM approach into a cotton production system implementing: (a) no-till soil management, (b) elimination/reduction of herbicides and insecticides, (c) full-time plant cover with both commodity and cover-crops accompanied with (d) introduction of a vermicompost sourced from Johnson-Su bioreactor injected into the furrow when planting promoted consistent year-over-year increases of 6.59 t C ha$^{-1}$ yr$^{-1}$ increase in SOC in the top 45 cm of the soil profile. This amount of SOC is ~6.6 times the estimates of the maximum technical potential established by *Lal et al. (2018)* of 1.0 t C ha$^{-1}$ yr$^{-1}$ or, up to ~33-times the range of observed SOC increases, ranging from 0.2 to 0.9 t C ha$^{-1}$ yr$^{-1}$ established by multiple meta-analyses of agroecosystem management research (*Ruis & Blanco-Canqui, 2017*; *Jian et al., 2021*; *Poeplau & Don, 2015*).

Increase in SOC and TSN values, observed in this research, are high; however, these increases occurred consecutively in each of the four years this research was conducted. While this amount of soil carbon accrual is intriguing, potential limitations may occur, depending on soil fertility, soil type, locale, environment, farm capabilities, weather, and rain/irrigation availability. The BEAM approach, in this field-scale test, warrants broader testing and replication is underway at other locations to confirm magnitude and repeatability and to assess the ultimate potential build-up and durability of the SOC.

These C-assimilation mechanisms were associated with an increase of TSN%, improving soil fertility and potentially reducing the need for future additions of synthetic N fertilizer. Also observed was the additional potential to offset another 8.87 t C ha$^{-1}$ yr$^{-1}$ through increasing cover crop residues, avoiding soil disturbance, and reducing (or eliminating) inputs: fertilizer, herbicides, insecticides, diesel, water, and labor. Together, the above listed benefits helped increased profitability and the likelihood for the future adoption of BEAM practices by farm managers.

*Mason et al. (2023)* concluded that some microbes possess the ability to influence soil C sequestration and retention and their ability may be utilized and promoted *via* inoculation of soil systems; however, their research emphasis was focused on single species organisms and they recommended that further research is required to assess potential candidates. This research implemented the injection of 2.2 kg ha$^{-1}$ of vermicompost extract, (made in a Johnson-Su bioreactor) as a multi-species microbial inoculum, into the furrow at plantings, of cover and commodity (cotton) crops. The injection of vermicompost appears to be the most likely "systems management" component that stimulated the observed increase of SOC% and TSN% when compared to the results of other research efforts.

Our knowledge of the soil microbiome is still in its infancy, due to the complexity of the community members and the diversity of the individual and communal interactions inherent in these systems. Increasing our understanding, of the structural and biological mechanisms of SMC in natural ecosystems for plant and soil health and fertility, nutrient acquisition, C exchanges, and plant and SMC carbon-use efficiencies will help us determine the potential negative and/or positive contributions of "transplanted" soil microbes to land-atmosphere C exchange and terrestrial C cycle climate feedbacks. Further research is now being pursued, to better understand the structure and composition of the soil microbiome, and the possible underlying mechanisms for carbon accrual and stabilization, and the significant increases in system TSN%.

The adoption of regenerative BEAM practices, in agroecosystems, may help us begin addressing other negative impacts conventional agriculture management is having on our agroecosystems, and the environment by: (a) avoiding the loss of soil C reserves, resulting from excessive soil disturbance from ripping, tilling and discing, (b) preventing further pollution of our soils, atmosphere, aquifers, rivers, estuaries and oceans, through elimination or reduction of applications of synthetic fertilizer, nitrogen (N), phosphorus (P), herbicides and pesticides, (c) decreasing the loss of topsoil from wind and water erosion, by promoting full-time ground cover, and (d) increasing top soil, SOC, TSN and microbiological resources.

These results are dependent on the adoption of a systems approach. The results of each practice in-and-of themselves, are not additive, and will likely not provide similar results. Soil disturbance must be reduced or eliminated, to avoid degradation of the integrity and functionality of the soil microbiome. Full-time plant cover, either commodity crops or cover crops must be implemented to increase total NPP, in order to fix an amount of carbon greater than SOC lost from herbivory and soil respiration. System nitrogen reserves must be increased through the activity of both free-living and symbiotic N fixing bacteria. The addition of a beneficial soil microbial community, from either a Johnson-Su vermicompost, or a vermicompost with a similar microbial community structure, to restore soil microbial community population, structure, diversity and system functionality will not fix the problems in the absence of implementing the other system components.

This small amount of vermicompost may be a temporary substitute or surrogate for the natural influence worms have on soil ecosystems. Increasing worm populations will require all of the above: (a) undisturbed soil conditions (limited-to-no plowing or discing), (b) no-or-reduced toxic agrichemicals/fertilizers, and (c) access to adequate plant biomass and plant root exudates to provide a soil environment with sufficient nutrient resources for earthworms to survive and reproduce. All these components work synergistically, as a "system" to facilitate a sustainable, productive and profitable agroecosystem.

The BEAM approach demonstrates a clean technology and employs sustainable, cost-effective practices. Adoption of BEAM brings both collective and individual expertise and resources, into agroecosystems that already have the equipment, infrastructure and personnel available. BEAM presents a feasible, scalable, self-sustaining and profitable solution for providing an immediate global impact on soil restoration and soil fertility,

while also productively utilizing atmospheric $CO_2$ as a necessary nutrient, and not as a waste to be discarded.

## ACKNOWLEDGEMENTS

We would also like to express our gratitude to Claire Bergamo, Debra Guo, and Juliet Russell; for their passion, intellect and investment of time, energy and effort to ensure all necessary resources, including timely communication and expert consultation, were available for successful integration of the BEAM technology at their participating farm in Turkey; and to the team members at SÖKTAŞ, a specialist maker of cotton and cotton-blended fabrics in Turkey: Muzaffer Kahan, Chief Executive Officer; Seyhan Aktemur, Chief Commercial Officer; and the Farm Managers and Operators: Basak Erdem, Barut Yasi, Hamdi Erdem, Seam Olmec; for, without these team members' commitment, integrity and farming expertise added to this project…these results would not have been possible.

### Funding

This work was funded by Stella McCartney, designer and manufacturer for luxury fashion; in conjunction with the Globetrotter Foundation/#NoRegrets Initiative. The funders had no role in study design, data collection and analysis, decision to publish, or preparation of the manuscript.

### Grant Disclosures

The following grant information was disclosed by the authors:
Stella McCartney, Designer and Manufacturer for Luxury Fashion.
Globetrotter Foundation/#NoRegrets Initiative.

### Competing Interests

The authors declare that they have no competing interests. All the information on the Johnson-Su bioreactor and the BEAM approach is open-source and freely available on the internet. The authors do not engage in consultancy activities, but do conduct research on the BEAM approach with grant support from foundations under the umbrella of the David C. Johnson LLC.

### Author Contributions

- David C. Johnson conceived and designed the experiments, performed the experiments, analyzed the data, prepared figures and/or tables, authored or reviewed drafts of the article, and approved the final draft.
- Hui-Chun Su Johnson conceived and designed the experiments, performed the experiments, analyzed the data, authored or reviewed drafts of the article, and approved the final draft.

## Field Study Permissions

The following information was supplied relating to field study approvals (*i.e.*, approving body and any reference numbers):

SOKTAS, were owners of the plot of land the research was conducted on and they requested the research be done on their fields.

## Data Availability

The raw data are available in the Supplemental File.

## Supplemental Information

Supplemental information for this article can be found online at http://dx.doi.org/10.7717/peerj.19167#supplemental-information.

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
