# Peer review of "Adoption of a biologically-enhanced agricultural management (BEAM) approach in agroecosystems for regenerating soil fertility, improving farm profitability and achieving productive utilization of atmospheric CO2"

_PeerJ, doi:10.7717/peerj.19167_

## Round 0.1 · original submission · Major Revisions

Please address the reviewers' comments in detail, especially the lack of replication.

Reviewer 1 ·

Basic reporting

I reviewed the manuscript in it's earlier version, and would conclude that in spite of the major improvement in including the bulk density change, the main problems are still present:
- the mechanism of C and N increase is not explained. The amount of additional C and N entering the system is very large, potentially unrealistic. Have similar C and N increases been found in other studies in such a climate?
- the experimental design is unclear and it is uncertain if the amount of samples can reliably be used to track the kinds of C changes observed. The authors avoid the discussion on the annual large fluctuation in samples.
- the scope of the manuscript is very broad and the introduction spans 8 pages, the discussion does not discuss the limitations and compares to other studies very broadly and the conclusions does not as much conclude the answers to research questions but speculate on the cost-efficiency of the practice.

Now with the bulk density (which still does not include the method in detail), an additional problem is presented: as the bulk density in the end of the experiment is less than in the beginning of the experiment, the sampled soil masses are not equal. The latter sampling took a shallower fraction of the soil, as the soil porosity improved. If the soil is stratified, then this shallower sampling in itself results in an increased TOC-% result. This was a common problem in no-till experiments historically and is a reason, why the equivalent soil mass approach is currently recommended.

Experimental design

Missing a control treatment. Unclear description of the experiment. Six composite samples, three N treatments + BEAM/non beam?
Not clear on the carbon inputs vs. C stock change.
Large interannual variation.
Large increases in C and N stocks, without a clear source.

Validity of the findings

The C and N increases seem incredibly high. The experiment does not seem to be properly replicated.

Reviewer 2 ·

Basic reporting

This article has a good introduction and background information that lead into the rest of the paper well. Overall, the grammar and English are professional. This section is a pass.

Experimental design

This study is not designed well. There are no replicates of the treatments and no clear way to discern if they controlled for or tested for any heterogeneity in the field. A randomized complete block design (with replicates) would have been a better design. They also draw conclusions about the impact the Johnson-Su bioreactor has on SOC accrual but this is an inappropriate conclusion without the proper control plot (no vermicompost).

Line 465 – are the farm mangers just “observing” this many earthworms or are they actually quantifying them? The word observation implies an estimate so if there is a robust method for quantifying them then that should be explained and more precise language should be used.

Validity of the findings

Line 544 – this conclusion paragraph is misleading. Yes a management difference between this study and the meta-analysis is that it used the vermicompost but without a direct control in this specific study that compares these results to C accrual without the vermicompost this is an inappropriate conclusion to make. Combined with a poor experimental design that does not replicate the treatments this interpretation of the results should not be made. This paper had done well up to this point appreciating the systems approach and understanding that the interactions/benefits of regenerative management are complex and not easily disentangled but if the goal was to show the benefit of the vermicompost specifically then the design should have included a “no vermicompost control.”

Line 562 – Are you trying to say that earthworms were the primary source of carbon sequestration? There is no data in this study that supports that mechanism so this whole section seems like a tangent with no relevant data besides earthworm counts. If this was the primary C sequestration pathway then it would follow that the different soil depths would have comparable SOC increases but that is not the case. It would be more beneficial to propose a mechanism that also explains the decrease in SOC in 15-30 cm or just not propose a mechanism at all.

Figure 4 – it is unclear how this figure is not double counting some of the carbon. There is SOC gained but then there is also cover crop biomass and no-till offsets, are those not part of the accrued SOC? The paragraph from line 623-631 should be more clear about the math in this figure and the descriptions do not clarify how the C is not being double counted. There seems to be assumptions that most of cover crop biomass is incorporated into the SOC when the research on this is very convoluted but most seem to agree that a majority of CC biomass on the surface is mineralized into CO2. The phrasing of carbon avoided/assimilated seems very greenwashy and the conclusions here are not well supported by results from the study/make assumptions about C-emissions vs sequestration and are mostly pulled from other sources and actual results.

Table S-3 – why is there a value for soil N increases each year? This is a hypothetical value and not a true income or expense. This is double counting the value of this N because the true value is in the crop yield (income) while applying less nitrogen (expense). What about other pesticides? It says weed control was 100% reduced but there were still insecticide costs that are not in this table so where is that cost accounted for.

Line 476 – this anecdote says soil aggregate stability was increasing? Is this measured somehow in the study? Also pretty impressive that this farmer can observe aggregate stability increasing without actually measuring it (yes that is sarcasm). Seems too specific to be believable and would be better if this was actually measured. Use the SLAKE app for cheap, easy testing.

·

Basic reporting

The paper is overall well written, clear, understandable language. Note it is missing “)” on line 475. Context for the project is clear and justified in both where and how the study was done and the focus on BEAM, SOC, and TSN. Authors make it clear that this is a systems approach and are evaluating inputs and outputs not explaining every process responsible for the results (particularly the soil fauna). The references are relevant and mostly recent throughout the paper and figures are clear, relevant, and well labelled.

Experimental design

Authors clearly stated question and justification for related research. The question exploring BEAM and N inputs is well defined and with adequate context for regenerative agriculture, C sequestration, and overall improved carbon footprint. The paper fills a gap in knowledge particularly related to necessity of N inputs alongside BEAM practices and impacts on soil from N additions on SOC and C storage. I appreciate that a more holistic approach to cost is used and detailed to include factors beyond just yield but recognize that this makes the design difficult in controlling for each variable. While not highly scientific, I like that farm manager observations are included as it’s a key component in a farm system. Methods are well organized between paper and supplemental materials. Specifically appreciate that the authors used more than one recommendation for N rates. I am unfamiliar with this compost method being referred to as vermicompost and have some additional questions about field design and number of samples taken for analysis.

Validity of the findings

These findings are beneficial in that they consider multiple factors in one study. This provides a more holistic and realistic understanding of the system than single components alone. These benefits could be meaningful to a variety of practitioners as they address inputs, yield, management, and C storage. It is important that there were multiple years of data collected to develop a robust and reliable data set regarding the SOC, TSN, and yields. While I think the various depths of sampling is very relevant, I think replicate samples would have been a valued addition. I have some question related to the long-term C storage which is commonly noted as a concern in agricultural systems. Long-term stability is mentioned but not really discussed explicitly. The conclusions are clearly stated related to yield and C storage, and they directly link to the feasibility for implementation on farm. Some of the extrapolated impacts throughout the paper seem to be a bit high based on other studies, but I recognize that this is a field study with such results.

Additional comments

Overall, this paper considers a number of components including yield, inputs, SOC, TSN, and management practices while still remaining thorough and acknowledging areas of limitation like fully understanding the microbiome communities. The paper is quite lengthy, particularly in the introduction.

---

## Round 0.2 · Minor Revisions

The authors need to include all suggestions given by the reviewers.

·

Basic reporting

I reviewed the manuscript in its previous state and think that most of the edits were made. I still feel that the introduction is lengthy and covers a lot of topics but I also recognize that this is a systems approach and covering many aspects of the system.

Experimental design

This is a field study and systems approach. A control would have been nice to account for more variables such as weather, however this is set up as a comparison over time. In a field study, I feel that it is sufficient. Detail has been added to address sampling scheme and compost as I previously commented. Limitations of the experiment have been added, which are an important addition and leave room for future work.

Validity of the findings

Some of the findings still seem quite high compared to other literature, but conclusions are drawn based on the 4 year field results. While I would be interested to see future work and replication, that is not the role or purpose of this paper. Some of the claims have been adjusted to more appropriately fit results of this project with less extrapolation. I appreciate that since this is a systems approach it includes impacts for farmers and land managers as they are part of this system.

Reviewer 4 ·

Basic reporting

This article describes a promising method for eco-friendly cotton production using a "Biologically-Enhanced Agricultural Management" (BEAM) system that saw an annual increase in soil carbon of 1.37 tons per hectare, an annual increase in total soil nitrogen of 0.68 tons/hectare, and an decrease in soil bulk density relative to conventionally managed plots within the same experiment. These increases accompanied increases in farm profits relative to parallel, traditionally managed plots. Profitability was largely attributed to reduced input costs in the BEAM system. The BEAM system integrated no-till practices, reduced or eliminated synthetic fertilizer applications, full time plant cover achieved by rotating cash crops with off-season multi-species cover crops, and application of vermicompost extracts.

This article provides a high value demonstration of a cotton production method that increases grower profits while reducing the environmental impacts of production. Such simultaneous ecological and economic gains are sought by many in regenerative agriculture, yet few studies integrate the same spectrum of approaches used here, and few studies document such clear beneficial outcomes.

The article reads clear, with the exception of several sentences in which the overuse of commas and parentheses is distracting. For example, the first sentence under "Background" reads, "A four-year field study, on the adoption of a Biologically-Enhanced Agricultural Management protocol, in a cotton/cover crop rotation in Turkey, was designed to observe "change-over-time" of soil organic carbon (SOC%) and total soil nitrogen (TSN%) at three soil profile depths (0-15 cm, 15-30 cm, and 30-45 cm) while tracking farm productivity and profitability." No commas are necessary in this sentence. Also, the three soil depths could be omitted, since these will be included in the methods section. The pattern of excessive comma usage is repeated throughout the paper. There are also many instances where words are capitalized that are not proper nouns. The manuscript would benefit from an English grammar editor to assist with a final revision.

The hypotheses tested were that the BEAM method would promote soil carbon and nitrogen levels, increase soil fertility and productivity, and increase farm profitability. Productivity was not explicitly defined.

The clear evidence of carbon and nitrogen accumulation, combined with profitability to the farmer observed using this integrated system has high value to producers and researchers in and beyond cotton production.

Experimental design

The experimental design was appropriate.

Validity of the findings

The statistical analysis utilized an online statistics platform, Statistics Kingdom, for analysis. I've never seen this approach used in a research paper intended for publication, so there is room to question whether the tool is adequate. Has it been used in other peer reviewed studies?

Statistics Kingdom algorithms seem to be based on XLS. The Statistics Kingdom website claims to compare their results to results obtained using R. If so, their algorithms shoud not be a concern. I would be curious to know what the consensus among reviewers is on this choice of statistical software, since a critical review of Statistics Kingdom is beyond the scope of what I can contribute to this review in the time allowed.

I am inclined to give the researchers the benefit of the doubt since their results, including the reports of increased profitability without remarkable yield gains is consistant with what I have seen in other crops when integrated methods that support soil health and biodiversity are implemented.

The profitability demonstrated in this study is no doubt influenced by local conditions at the time of the study. The authors note the need for additional research. Clearly other crops, ecosystems, and markets all need to be analyzed. However, the benefits of the BEAM approach are consistant with benefits noted by research cited within the paper, and benefits widely promoted by soil health experts. What makes this study unique is the decision to include profitability measures from an established producer, since there is little incentive for growers to adopt ecological practices that lead to financial losses.

Additional comments

The review questionaire asks whether all required permits were used. As a reviewer with no experience in Turkey, I know nothing of the required field permits.
The researchers had permission from the land owners, and included the name of the approving organization in their manuscript.

I have included an annotated copy of the manuscript. Among the annotations were comments about images I did not initially find. I found them later, so the remarks can be disregarded.

Annotated reviews are not available for download in order to protect the identity of reviewers who chose to remain anonymous.

---

## Round 0.3 · accepted · Accept

The revised manuscript is good to accept.